# Translational Potential of Metabolomics on Animal Models of Inflammatory Bowel Disease—A Systematic Critical Review

**DOI:** 10.3390/ijms21113856

**Published:** 2020-05-29

**Authors:** Lina Almind Knudsen, Rasmus Desdorf, Sören Möller, Signe Bek Sørensen, Axel Kornerup Hansen, Vibeke Andersen

**Affiliations:** 1Focused Research Unit for Molecular Diagnostic and Clinical Research, University Hospital of Southern Denmark, Kresten Philipsens Vej 15, 6200 Aabenraa, Denmark; lina.almind.knudsen@rsyd.dk (L.A.K.); rasmus.desdorf@gmail.com (R.D.); Signe.Bek.Sorensen@rsyd.dk (S.B.S.); 2IRS-Center Soenderjylland, University of Southern Denmark, Winsloewparken 19, 3, 5000 Odense C, Denmark; 3OPEN—Odense Patient data Explorative Network, Odense University Hospital, J. B. Winsloews Vej 9a, 5000 Odense C, Denmark; Soren.Moller@rsyd.dk; 4Department of Clinical Research, University of Southern Denmark, J. B. Winsloews Vej 19, 5000 Odense C, Denmark; 5Department of Molecular Medicine, University of Southern Denmark, J. B. Winsloews Vej 21-25, 5000 Odense C, Denmark; 6Department of Veterinary and Animal Sciences, Faculty of Health and Medical Sciences, University of Copenhagen, Groennegaardsvej 15, 1870 Frederiksberg C, Denmark; akh@sund.ku.dk

**Keywords:** inflammatory bowel disease, metabolomics, animal models, systematic review

## Abstract

In the development of inflammatory bowel disease (IBD), the gut microbiota has been established as a key factor. Recently, metabolomics has become important for understanding the functional relevance of gut microbial changes in disease. Animal models for IBD enable the study of factors involved in disease development. However, results from animal studies may not represent the human situation. The aim of this study was to investigate whether results from metabolomics studies on animal models for IBD were similar to those from studies on IBD patients. Medline and Embase were searched for relevant studies up to May 2017. The Covidence systematic review software was used for study screening, and quality assessment was conducted for all included studies. Data showed a convergence of ~17% for metabolites differentiated between IBD and controls in human and animal studies with amino acids being the most differentiated metabolite subclass. The acute dextran sodium sulfate model appeared as a good model for analysis of systemic metabolites in IBD, but analytical platform, age, and biological sample type did not show clear correlations with any significant metabolites. In conclusion, this systematic review highlights the variation in metabolomics results, and emphasizes the importance of expanding the applied detection methods to ensure greater coverage and convergence between the various different patient phenotypes and animal models of inflammatory bowel disease.

## 1. Introduction

Inflammatory bowel diseases (IBDs) are chronic, relapsing disorders of the gut, comprised mainly of Crohn’s disease (CD) and ulcerative colitis (UC) [1]. The inflammation in CD is transmural and patchy and can affect the entire gastrointestinal tract, while UC is confined to the colon and primarily involves the mucosa in a continuous manner [2]. The underlying disease mechanisms in IBD are still being uncovered, but the etiology is known to be multifactorial and governed by host genetics and environmental factors including the gut microbiota [3]. Discovering details about the development of IBD has been aided in recent years by new and improved methods to detect and quantify factors believed or known to be involved in these diseases. For instance, novel sequencing methods have made it possible to study genome variations and the microbiota in greater detail through metagenomics, while proteomics now has greater molecular coverage and improved quantification accuracy. As IBD has a multifactorial etiology and affects the system on multiple levels simultaneously, several complementary analyses are needed to reveal the underlying pathological mechanisms. One of the newest “-omics” applied to this field is metabolomics. In this respect, metabolomics can be considered as a functional analysis investigating metabolites resulting from metabolic processes and thereby add to the “static” genetic analyses. Indeed, the detection and quantification of metabolites have revealed metabolites that allow discrimination between IBD patients and healthy controls [4,5]. It is also a supplement to microbiota sequencing, when trying to understand the functional relevance of disease-related changes in the microbiota. For instance, bacteria that produce short-chain fatty acids (SCFAs) are reduced in feces from IBD patients [6,7], while *Card9^-/-^* mice, which are more susceptible to colitis, have an altered microbiota unable to metabolize tryptophan [8]. The effect of these microbiota changes can be investigated using metabolomics, thus potentially making metabolomics a key factor in discovering diagnostic biomarkers and understanding the role of microbiota and dysbiosis in the development of IBD.

Animal models are an invaluable tool for discovery and have provided valuable insights into various disease mechanisms [9]. However, with the emergence of high-throughput omics technologies, further details on mechanistic insights are within reach. Metagenomics has been applied to animal models, including animal models for IBD [10,11]. This method will help elucidate the important interactions between the gut microbiota and the development of multiple diseases, which is needed for a better understanding of disease pathogenesis and the development of new treatment strategies. Metabolomics is still a developing method, and therefore little is known about the translational value of these data. In this review, we have compared metabolomic findings in animal models of IBD and IBD patients, in order to evaluate the translational potential of metabolomics data found in animal models of IBD. The aims of the review were to (1) identify metabolites differentiated between IBD cases versus healthy controls in both animal models and humans, (2) investigate correlations between different key experimental elements and specific metabolites, and (3) determine if the metabolome of a specific animal model is representative of the metabolome of IBD or an IBD subtype in humans. Data showed a convergence of ~17% for metabolites differentiated between IBD and controls in human and animal studies, and the dextran sodium sulfate model appeared as a good model for analysis of systemic metabolites in IBD. Other key experimental elements did not show clear correlations with any significant metabolites.

## 2. Results

### 2.1. Study Characteristics

Fifty-eight studies met our search criteria and were included in this review (Figure 1), of which 32 were human studies, 25 were animal model studies, and one study presented data from both humans and an animal model. The human studies were categorized according to disease (CD, UC, IBD) and age, while the animal model studies were categorized according to model type and age of the animals (Table 1). If animals in a study were grouped spanning more than one age group, the study was characterized according to the older age group. Descriptive characteristics for all studies were extracted, with different tables for the human and animal studies, respectively (Appendix A).

### 2.2. Quality Assessment

Two sets of quality criteria were used to assess the quality of the human and animal studies, respectively (Appendix A). Each study was assigned as being of “good”, “medium”, or “poor” quality, based on the amount of quality criteria fulfilled, as presented in Table 2. The majority of studies (75%) were of medium quality, while only 9% of all studies were considered good.

### 2.3. Metabolites Differentiated in Inflammatory Bowel Disease (IBD) Cases Versus Healthy Controls in Both Humans and Animal Models

A total of 200 different metabolites were reported as being increased in IBD across all included human studies, while 218 were decreased (Table 3). The numbers were higher for the animal studies with a total of 280 different metabolites reported as being increased in IBD, while 253 were decreased. Some metabolites were reported as both increased and decreased in each study type, but the majority was exclusively reported as increased or decreased. Results for human and animal model studies, respectively, are presented in separate tables for metabolites that are increased and decreased in each type of study (Appendix A).

To assess the similarities in metabolomics findings between study types, metabolites increased or decreased in IBD in both human and animal studies were identified and are presented in Table 4; Table 5. Forty-eight metabolites were found to be increased in both types of studies, while 41 metabolites were decreased. This corresponds to 17% of metabolites found increased and 16% of metabolites found decreased in IBD in animal studies also being reported as increased and decreased, respectively, in human IBD studies. Of this subgroup of metabolites, 21 were reported as both increased and decreased, respectively, in IBD including several amino acids, and this overlap can largely be explained by the variation in study details. This leaves 27 metabolites exclusively increased, and 20 metabolites exclusively decreased in IBD in both human and animal studies (in bold in Table 4 and Table 5).

### 2.4. Metabolites of Special Interest

Several tryptophan metabolites were found to be regulated in human studies, animal studies, or both. Kynurenine and quinolinic acid were increased in UC and CD patients, respectively (Appendix A). Kynurenine was also found to be increased in DSS (dextran sodium sulfate) and IL-10^-/-^ mouse models (Appendix A), while quinolinic acid was decreased in IL-10^-/-^ mice along with kynurenic acid and 5-hydroxyindoleacetic acid (Appendix A). Additionally, 5-hydroxytryptophan and 3-hydroxykynurenine were also increased in DSS and IL-10^-/-^ mouse models, respectively (Appendix A). Conflicting observations were made for tryptophan itself, which was reported to be both increased and decreased in human studies as well as the DSS mouse model (see Table 4 and Table 5). SCFAs were reported to be regulated in numerous human IBD studies, although some results were conflicting. Formic acid and acetic acid were thus observed to be both increased and decreased in CD and UC patients, depending on the study (Appendix A). However, propionic acid, butanoic acid, isobutyric acid, and pentanoic acid were all observed to be decreased in CD and UC patients (Appendix A). Interestingly, only animal studies using the acute DSS mouse model or the TNBS (2,4,6-trinitrobenzenesulfonic acid) rat model reported differentiated levels of SCFAs (Appendix A). Acetic acid was decreased in the DSS model, while butanoic acid was decreased in the TNBS model (Appendix A). Dong et al. [13] also observed butanoic acid to be decreased, but only on the first day of DSS, after which it was increased throughout the experiment.

### 2.5. Included Studies Are Characterized by Great Variation in the Key Experimental Elements

A metabolomics study consists of several different key experimental elements that can vary between studies. Here, these elements are the experimental subjects (disease subtype for the human studies and species, strain, and type of model for the animal studies), biological sample type, analysis methodology, and age of experimental subjects/study population. Large variations in these elements can make it difficult to compare results across the different studies and thereby difficult to draw any overall assumptions on the topic in question.

To clearly elucidate the large variation between the different studies included in this review, we tallied up the number of studies containing the different variants of each key experimental element in animal studies and human studies, respectively (see Table 6 and Table 7). Looking at Table 6 and Table 7, it becomes immediately clear that there could be a very high degree of variation between studies as a result of the different elements applied in the studies. For the animal studies (Table 6), three different species with a total of 11 different mouse and rat strains were used along with eight different IBD animal models, three main analytical platforms, 13 different sample types, and four different age groups across the 26 studies. The variation in study population and sample type was less for the human studies (Table 7), however seven different analytical platforms were applied, giving rise to a considerable heterogeneity across the human studies.

A few studies did, however, share a high degree of similarity in experimental factors. Animal studies by Shiomi et al., Gu et al., and Wang et al. all used C57BL/6J mice from the same age group for a 3% DSS model as well as using gas chromatography-mass spectrometry (GC-MS) to detect metabolites in serum and colon samples (see Appendix A) [17,37,55], although it is worth noting that Gu et al. and Wang et al. belong to the same department at Kobe University, Japan. Equally, two studies by the same first author also shared a similar degree of similarity using an IL10^-/-^ model [33,40]. For the human studies, two studies used proton nuclear magnetic resonance (^1^H-NMR) to detect metabolites in serum samples from CD and UC patients of 18-60+ years of age [15,43], while two other studies detected metabolites in serum samples from CD and UC patients in the >1–60+ age groups using GC-MS [20,24]. The authors of the latter two studies are also from the same department and even co-authors the other study, again underlining the difficulties at present comparing studies from different research groups.

### 2.6. Differentiation of Metabolites According to Key Experimental Elements 

We found that in both human and animal studies, the vast majority of the metabolites were detected by more than one analytical platform (Appendix A). The study subjects in most of the human studies spanned all age groups from very early onset and young to old, making it difficult to differentiate metabolite detection between age groups in the human studies. However, most metabolites were generally detected in more than one age group in the animal studies, suggesting that age is not a deciding factor when it comes to the metabolome. Nevertheless the amino acid isoleucine stood out, as it was increased only in human subjects above 18 years of age and in mice of >8–24 weeks. One of the animal studies that reported increased levels of isoleucine also included animals of 1 week, but the amino acid was not significantly altered in this group [26].

The subgroup of metabolites differentiated in both study types was sorted according to the biological sample types in which they were detected (Appendix A). This allowed us to examine any parallels between human and animal studies. Many metabolites were found in several different sample types in both humans and animals, but not necessarily the same. For example, alanine was increased in serum [24] and feces [4,25] from humans and in colon [17] and plasma [26] from mice, but it was decreased in urine [43] and colon [35,36] in humans and serum [16] and urine [38] in animals, illustrating the differences observed for many metabolites (Appendix A). The highest similarity to human studies was observed with the acute DSS mouse model (Appendix A). Since this model was used in almost half of the included animal studies, this finding is not surprising. However, only five of the acute DSS mouse model studies analyzed serum samples, but still 11 of the increased and 11 of the decreased metabolites were detected in serum samples from both humans and the DSS mouse model. A total of 34 and 29 different metabolites were reported as increased and decreased in IBD, respectively, in serum samples from the acute DSS mouse model. This means that 32% of the increased metabolites and 38% of the decreased metabolites in serum samples from the acute DSS mouse model were reported to be correspondingly differentiated in the human studies. Conversely, the acute DSS mouse model could account for 16% (22 out of 136 metabolites) of the overall metabolite changes observed in serum of IBD patients. This could suggest serum samples from the acute DSS mouse model as having good translational potential when analyzing systemic metabolites in IBD.

### 2.7. Correlation between Animal Models and IBD Subtypes

For all the metabolites significantly differing in both human and animal studies, it was investigated if some animal models were specifically good models for CD or UC when it comes to metabolomics (Appendix A). Most of the models had similarities with both CD and UC. For instance, regarding metabolites decreased in the IL10^−/–^ mouse model, glucose was also decreased in CD, while leucine was decreased in UC, and trimethylamine in both CD and UC. The TNF^ΔARE/WT^ model only had similarities with UC, but this could easily be due to the fact that only one study with this model was included. Overall, this indicates that the metabolomes of the animal models included in this review are not correlated specifically to CD or UC.

### 2.8. Metabolite Classifications

All metabolites differentiated between IBD cases and controls in either humans or animals were sorted into metabolite subclasses according to the classification system used in The Human Metabolome Database (www.hmdb.ca) (Appendix A). The most differentiated subclass was “amino acids, peptides, and analogues” in both human and animal studies, representing approximately 16% of all differentiated metabolites reported. “Fatty acids and conjugates” as well as “carbohydrates and carbohydrate conjugates” were also among the most differentiated in both human and animal study types. “Glycerophosphocholines” were also differentiated in both, but to a much larger extent in animal studies. In general, different kinds of lipids were reported more frequently as differentiated in IBD in animal studies compared to human studies. Metabolites from 142 different subclasses were reported as differentiated between IBD and controls overall. Of these, 47 were differentiated in both human and animal studies, while 48 and 47 differentiated subclasses were unique to human and animal studies, respectively. This shows a large gap between the type of metabolites that are investigated and detected in the two study types, as only a third of the total amount of differentiated subclasses are reported in both.

When focusing on the metabolites differentiated in IBD in both human and animal studies, they represented a total of 25 subclasses overall. Metabolites from nine different subclasses were present among both the increased and decreased metabolites, while eight subclasses were exclusively increased and decreased, respectively.

## 3. Discussion

This systematic review was conducted to assess the overall translational value of conducting metabolomics analyses on animal models of IBD. We found that approximately 17% and 16% of metabolites reported as differentiated between IBD and controls in the animal studies were also reported as differentiated in human studies for both increased and decreased metabolites, respectively. Amino acids accounted for almost half of these metabolites. Tryptophan metabolites were differentiated in both human and animal studies, and reduced tryptophan metabolism has been associated with colitis [8]. Here, two metabolites of the kynurenine pathway were observed to be increased in IBD patients, while one from the kynurenine pathway and one from the serotonin pathway were increased in the acute DSS mouse model. In the IL-10^-/-^ mouse, two metabolites from the kynurenine pathway were increased and two were decreased, while another was decreased in the serotonin pathway. SCFAs were also differentiated in both human and animal studies. Most results showed decreased levels of SCFAs, which corresponds well to the reduced levels of SCFA-producing bacteria seen in IBD patients [6,7]. Interestingly, animal studies using the acute DSS or TNBS models reported differentiated levels of SCFAs. Data from metabolomics on serum samples from the acute DSS mouse model showed a convergence of 29% for increased and 38% for decreased metabolites in human serum samples. All this is indicative of a good translational potential of the acute DSS mouse model. However, results from different animal models were not correlated specifically to CD or UC.

In the animal studies included in this review, the majority of the studies used models induced as acute models, which is in contrast to the fact that IBD in the human patients is a chronic condition [1]. However, the 68 different metabolites differentiated in both IBD patients and IBD animal models were from both acute and chronic models. This, along with the observation that the acute DSS model had the most similarities with IBD patients in terms of differentiated metabolites, indicates that the specific mechanism operative in the genesis of the inflammation may be of greater importance than whether an animal model is induced as acute or chronic when studying metabolomics in IBD [56]. It should, however, be noted that none of the existing models truly recapitulates the spontaneous and fluctuating nature of the human disease. The limitations of each model should always be taken into consideration before directly applying experimental findings to the human condition [57].

Only a few of the human studies provided information on the clinical phenotype. Some studies provided information about localization of disease, but only a few stratified for this in their results, although studies have shown a correlation between disease phenotype and gut microbiota composition in CD [58,59]. Different animal models would be expected to explain different phenotypical traits of IBD. In order to uncover these associations, it is essential that the phenotype of human IBD subjects is described in greater detail with more clinical information regarding e.g., disease localization and the course of the disease.

The different analytical techniques used to detect metabolites, mainly nuclear magnetic resonance (NMR) and mass spectrometry (MS), each have their strengths and weaknesses, and no technique is able to completely identify and quantify all metabolites within a sample [60,61,62]. This could explain some of the large variation observed in the metabolites detected and reported as differentiated in IBD in human and animal studies, respectively, because several different techniques were employed. However, we still find that most of the metabolites differentiated in both human and animal studies were detected by at least two different techniques, showing some degree of agreement between the coverage of the different techniques after all. 

The majority of the human studies were conducted using GC-MS or NMR spectroscopy, while liquid chromatography-MS (LC-MS) was used more in the animal studies. NMR spectroscopy is quantitative and very accurate, but has a low sensitivity compared to MS [60]. GC-MS analysis provides good reproducibility, but is limited to the detection of volatile compounds, whereas LC-MS can be combined with various ionization techniques to optimize detection of specific classes of metabolites [60]. These different techniques are complementary and using more than one technique to analyze the same sample will increase coverage and provide a more complete representation of the metabolites within a sample. The total number of detected metabolites reported from especially the human studies could thus potentially have been increased by using a combination of techniques. This would increase the overall coverage and thereby also the possibility of seeing more similarities between different study types.

For the animal studies, age did not appear to be a deciding element for the metabolome in IBD models, as no (particular) difference in metabolite composition were seen between different age groups. In humans, we know that different microbiomes of the body, e.g., the gut and skin microbiomes, change with age [63,64,65], and the same has been found for the gut microbiota in mice and rats [66,67]. It has also been shown that blood metabolites are affected by the gut microbiota [68]. So, it is reasonable to speculate that metabolomes in the body should also change over time as a consequence of age-related changes in the gut microbiome. Indeed, a longevity study by Collino et al. [69] showed different metabolomic signatures in different age-groups between 24 and 111 years of age. Here, the lack of an age-related effect on the animal model metabolomes could be attributed to the low number of studies with infant, juvenile, or old animals (see Appendix A) compared to the number of studies with adult animals. More studies in each age group are possibly needed for this effect to become apparent.

As the study populations in most of the included human studies spanned all age groups, no conclusions could be drawn for metabolite differentiation between age groups. As for the animal studies, it would be of interest to have metabolomics studies investigating human study populations of specific ages, to be able to report on potential age-related differences in the metabolome. This would preferably be performed alongside microbiome analyses to study metabolome-microbiome associations across different age groups.

For the subgroup of metabolites both increased and decreased in both study types, a closer look at the data revealed that differences in disease activity, age, IBD subtype, and sample type could explain these apparently opposing results. This underlines the importance of including detailed information about the subject population, when evaluating research results.

In conclusion, the acute DSS model appeared to be the best animal model for metabolomics in IBD and could account for 16% of the metabolite changes seen in serum of IBD patients. The great variation in results between study types suggests that it is necessary to align and expand the choice of detection methods and biological sample types analyzed in order to be able to accurately compare metabolomics analyses performed in humans and animals. Furthermore, transdisciplinary research is needed to ensure results that can be translated for use in the clinical setting and benefit the patients.

## 4. Materials and Methods

### 4.1. Search Strategy

The databases searched (up to May 2017) were Embase Classic + Embase 1947 to 2017 (382 hits) and MedLine (via PubMed) (182 hits). The search was limited to English and Danish language manuscripts and using a combination of terms for (1) metabolomics, (2) mass spectrometry and spectroscopy, and (3) inflammatory bowel disease. Exact terms used for each group in each database can be found in Appendix A. A total of 560 hits were found, 151 of which were removed by EndNote as duplicates, leaving 409 hits for screening.

### 4.2. Selection Criteria

We wanted to include all patient studies and animal model studies evaluating differences in metabolites between IBD cases and healthy controls. Studies were excluded if they (1) were performed with animals other than mice, rats, or pigs; (2) did not have an appropriate control group (e.g., individuals with other gastrointestinal diseases); (3) did not show if differences in metabolites were significant; (4) were in a language other than English and Danish. The Covidence systematic review software (Veritas Health Innovation, Melbourne, Australia) was used for abstract and full-text screenings, the latter of which was performed independently by L.A.K and R.D., the former by L.A.K. only. In cases of discrepancies between the independent screenings, these were resolved by a discussion between the two screening authors. The flowchart of the study screening process in Covidence can be seen in Figure 1. In spite of a thorough literature search strategy, many irrelevant studies were still included in the search results. Of the 409 screened studies, 318 studies were excluded already during the abstract screening due to a number of reasons. More than half of the 318 excluded studies were reviews, editorials, or abstracts. A few were in a language other than English or Danish or duplicates not removed by the reference program. Others were on patients with other diseases than IBD (e.g., Clostridium difficile infection or necrotizing enterocolitis) or IBD animals with other conditions aside from IBD. Additionally, several studies focused solely on degradation products from IBD drugs, while some were in fact microbiome or genome-wide association studies with no metabolomics data. During the full-text screening, 10 studies were excluded with the reason “wrong outcomes” and 12 were excluded with the reason “wrong study design”. An example of a wrong outcome was prediction performance estimates, i.e., how well metabolomics could discriminate between UC and control without any quantitative data for specific metabolites. A study categorized as having a wrong study design had two separate studies—one with healthy volunteers and one with CD patients, without any comparison of healthy and CD. After the full-text screening, a total of 58 studies were selected for the analysis of metabolomics studies in animal models for IBD and IBD patients [4,5,6,13,14,15,16,17,18,19,20,21,22,23,24,25,26,27,28,29,30,31,32,33,34,35,36,37,38,39,40,41,42,43,44,45,46,47,48,49,50,51,52,53,54,55,70,71,72,73,74,75,76,77,78,79,80,81].

### 4.3. Data Extraction

All data were extracted using three checklists for patient studies and animal studies, respectively: descriptive, quality, and results. The descriptive checklist was used for population characteristics and technical details about metabolomics. The quality checklist for patient studies was based on the QUADOMICS tool for quality assessment [82], while that of the animal studies was based on both the QUADOMICS tool and the animal study specific SYRCLE [83]. Descriptive and quality checklists with data can be seen in the supplementary material (Appendix A). Aspects taken into consideration for the summarized animal study results were species, biological sample type, metabolomics platform, type of animal model, and references. For the human studies, similar aspects were included: disease (incl. location if specified), disease activity (active, inactive), biological sample type, age, metabolomics platform, and references. Sex was not included as all human studies analyzed both male and female IBD patients or found no differences in metabolite detection between males and females. Data were extracted by LAK and RD and assisted by SM on statistical matters to ensure correct evaluation of included studies.

The nomenclature in metabolomics is redundant in some cases (e.g., butyric acid and butanoic acid are two names for the same fatty acid, while butyrate and butanoate are the corresponding names for their conjugate bases), and we have attempted to minimize this redundancy by gathering identical metabolites under one name. In the case of acids, we have chosen to report the acid instead of the conjugate base.

PROSPERO Registration number: CRD42017068289

## Figures and Tables

**Figure 1 ijms-21-03856-f001:**
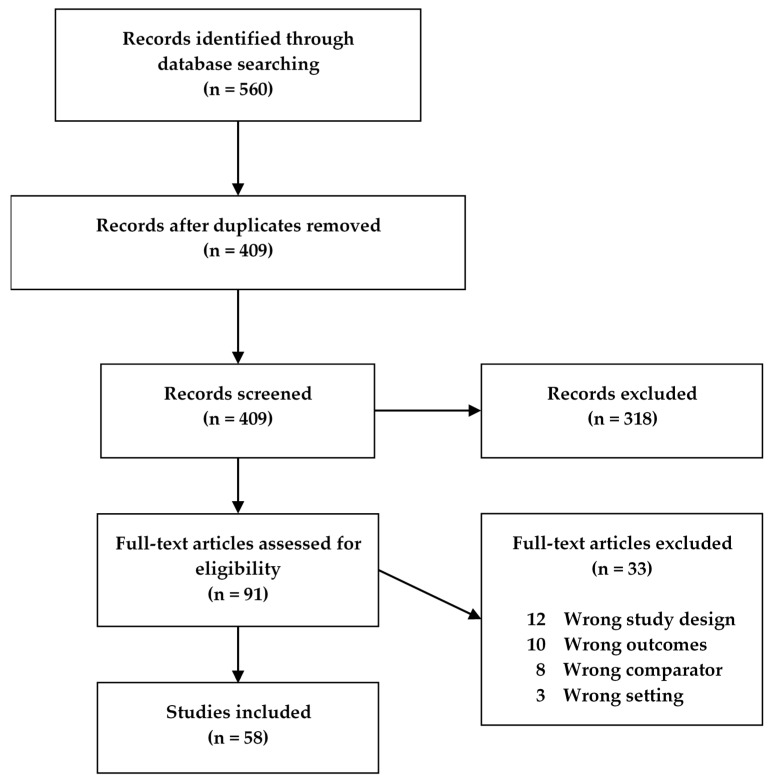
Flowchart of the study screening process for original studies in metabolomics for inflammatory bowel disease (IBD) patients and IBD animal models.

**Table 1 ijms-21-03856-t001:** Age categories for mouse studies (a) and human studies (b) in the systematic review on metabolomics in inflammatory bowel disease (IBD) patients and IBD animal models.

Mouse Studies		Human Studies	
Phase of Life	Age in Weeks	Phase of Life	Age (Years)
Infant	0–3	Infant	0–1
Juvenile	>3–8	Very early onset and young	>1 and <18
Adult	>8–24	Adult	18–60
Old	>24	Old	60+

(Modified from [12]).

**Table 2 ijms-21-03856-t002:** Quality assessment of studies included in the systematic review on metabolomics in inflammatory bowel disease (IBD) patients and IBD animal models.

Level of Quality	% of Criteria Fulfilled	Animal Studies	Human Studies	All Studies
**Good**	≥70%	12%	6%	9%
**Medium**	40–70%	69%	79%	75%
**Poor**	<40%	19%	15%	17%

**Table 3 ijms-21-03856-t003:** Number of differentiated metabolites detected across study types included in the systematic review on metabolomics in inflammatory bowel disease (IBD) patients and IBD animal models.

Number of Different Metabolites Detected	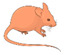	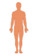	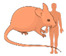
Animal Studies	Human Studies	Both
Increased	280	200	48	48/280 = 17%
Decreased	253	218	41	41/253 = 16%
Exclusively increased	215	135	27	
Exclusively decreased	190	153	20	

**Table 4 ijms-21-03856-t004:** Metabolites significantly increased in inflammatory bowel disease (IBD) vs healthy controls in both humans and animals in the systematic review.

	Human Studies						Animal Studies					
Metabolite *	Disease	Activity	Sample Type	Age Group	Platform	References	Species	Sample Type	Age (Weeks)	Platform	Model	References
**3-Hydroxybutyric**	UC, IBD	AC	Serum	A, O	^1^H NMR	[14,15]	Mouse	Serum	>3–8	^1^H NMR	DSS (A)	[16]
**acid**							Mouse	Serum	>8–24	GC-MS	DSS (A)	[17]
**4-Hydroxyphenyl-**	CD	AC	Urine	A, O	^1^H NMR	[18]	Mouse	Colon	>8–24	GC-MS	DSS (A)	[17]
**acetic acid**	CD, UC	All	Urine	Y	^1^H NMR	[19]						
**Acetoacetatic acid**	IBD	AC	Serum	A, O	^1^H NMR	[15]	Mouse	Serum	>3–8	^1^H NMR	DSS (A)	[16]
	IBD	IA	Urine	A, O	^1^H NMR	[15]						
**Acetylaspartic acid**	UC	All, AC, IA	Serum	Y, A, O	GC-MS	[20]	Mouse	Colon (distal), cecum	0–3	UPLC/Tof-MS	T-syn deficiency	[21]
Acetylcarnitine	CD, UC	AC	Urine	A, O	^1^H NMR	[18]	Mouse	Colon	>8–24	LC-qTOF-MS	DSS (C)	[22]
Acylcarnitine	CD	All	Urine	Y	^1^H NMR	[19]	Mouse	Ileum (distal)	>8–24	LC-MS	*TNF* ^ΔARE/WT^	[23]
Alanine	CD	All	Serum	Y, A, O	GC-MS	[24]	Mouse	Colon	>8–24	GC-MS	DSS (A)	[17]
	CD	Unknown	Feces	Y, A, O	^1^H NMR	[25]	Mouse	Plasma	>3–24	^1^H NMR	*IL10^-/-^*	[26]
	CD, UC	AC	Feces	A, O	^1^H NMR	[4]						
**Arachidonic acid**	CD (ICD)	IA	Feces	Y, A, O	FT-ICR-MS	[27]	Mouse	Ileum (distal)	>8–24	LC-MS	*TNF* ^ΔARE/WT^	[23]
							Mouse	Colon (distal), cecum	0–3	UPLC/Tof-MS	T-syn deficiency	[21]
**Arginine**	CD	AC	Plasma, serum	A, O	^1^H NMR	[18]	Mouse	Liver	>8–24	LC-qTOF-MS	DSS (C)	[22]
	UC	AC	Urine	A, O	^1^H NMR	[18]	Mouse	Plasma	>3–24	^1^H NMR	*IL10^-/-^*	[26]
**Butanal**	CD	All	Breath	A, O	SIFT-MS	[28]	Mouse	Feces	>8–24	GC-MS	*Winnie*	[29]
Carnitine	CD, UC	AC	Urine	A, O	^1^H NMR	[18]	Mouse	Colon	>8–24	LC-qTOF-MS	DSS (C)	[22]
**Cholic acid**	CD	IA	Feces	Y, Unknown	UPLC/ToFMS	[30]	Rat	Plasma	?	UPLC-ESI-QTOF-MS	TNBS	[31]
Creatine	CD	AC	Plasma	A, O	^1^H NMR	[18]	Mouse	Serum	>3–8	^1^H NMR	DSS (A)	[16]
	UC	AC	Plasma, serum	A, O	^1^H NMR	[18]	Mouse	Plasma	>3–8	^1^H NMR	*IL10^-/-^*	[26]
**Dimethylamine**	IBD	IA	Serum	A, O	^1^H NMR	[15]	Rat	Urine	?	UPLC-MS/MS	TNBS	[32]
**Ethylmalonic acid**	UC	All, AC, IA	Serum	Y, A, O	GC-MS	[20]	Mouse	Colon	>8–24	GC-MS	DSS (A)	[17]
**Fructose**	UC	IA	Serum	Y, A, O	GC-MS	[20]	Mouse	Feces	>8–24	GC-MS	*Winnie*	[29]
Fumaric acid	CD, UC	All	Serum	Y, A, O	GC-MS	[24]	Mouse	Urine	>8–24	^1^H NMR	DSS (A)	[13]
							Mouse	Plasma	>3–8	^1^H NMR	*IL10^-/-^*	[26]
Glucose	UC	AC	Serum	A, O	^1^H NMR	[14,18]	Mouse	Urine	>8–24	GC-MS	*IL10^-/-^*	[33]
	UC	All	Feces	A, O	^1^H NMR	[34]						
	UC	All, AC, IA	Serum	Y, A, O	GC-MS	[20]						
	UC	IA	Colon	Unknown	Proton MRS	[35]						
	CD, UC	AC	Colon	Unknown	Proton MRS	[35]						
	IBD	AC	Colon	A	^1^H NMR	[36]						
Glutamic acid	UC	Unknown	Feces	Y, A, O	^1^H NMR	[25]	Mouse	Colon	>8–24	GC-MS	DSS (A)	[17]
	UC	All, AC, IA	Serum	Y, A, O	GC-MS	[20]						
**Glycerol**	UC	AC	Serum	Y, A, O	GC-MS	[20]	Mouse	Plasma	>8–24	^1^H NMR	DSS (A)	[13]
	CD	AC	Plasma	A, O	^1^H NMR	[18]	Mouse	Feces	>8–24	GC-MS	*Winnie*	[29]
Glycine	CD	AC	Serum	A, O	^1^H NMR	[18]	Mouse	Colon	>8–24	GC-MS	DSS (A)	[37]
	CD	AC, IA	Feces	A, O	^1^H NMR	[4]	Mouse	Feces	>8–24	^1^H NMR	Adoptive	[38]
	CD, UC	All	Urine	Y	^1^H NMR	[19]					transfer	
	CD, UC	All	Serum	Y, A, O	GC-MS	[24]						
	IBD	AC	Serum	A, O	^1^H NMR	[15]						
**Hydroxybenzoic acid**	UC	All, AC	Serum	Y, A, O	GC-MS	[20]	Mouse	Colon, serum	>8–24	GC-MS	DSS (A)	[17]
Inositol	CD	AC	Feces	A	GC-MS	[18]	Mouse	Feces	>8–24	GC-MS	*Winnie*	[29]
Isoleucine	CD	AC	Serum	A	^1^H NMR	[39]	Mouse	Colon, serum	>8–24	GC-MS	DSS (A)	[17]
	CD	Unknown	Feces	Y, A, O	^1^H NMR	[25]	Mouse	Plasma	>8–24	^1^H NMR	*IL10^-/-^*	[26]
	CD, UC	AC	Feces	A, O	^1^H NMR	[4]	Mouse	Feces	>8–24	^1^H NMR	Adoptive	[38]
	CD, UC	AC	Serum, plasma	A, O	^1^H NMR	[18]					transfer	
	IBD	AC	Serum	A, O	^1^H NMR	[15]						
**Kynurenine**	UC	All, AC, IA	Serum	Y, A, O	GC-MS	[20]	Mouse	Plasma	>8–24	LC-MS	*IL10^-/-^*	[40]
							Mouse	Plasma	>8–24	UPLC-MS	DSS (A)	[41]
Lactic acid	CD	AC	Plasma, urine	A, O	^1^H NMR	[18]	Mouse	Colon	>8–24	NMR (^1^H, ^1^C, ^1^P)	DSS (A)	[42]
	UC	AC	Urine	A, O	^1^H NMR	[18]	Mouse	Plasma	>3–24	^1^H NMR	*IL10^-/-^*	[26]
	UC	AC	Feces	A, O	^1^H NMR	[4]						
	UC	All	Urine	Y	^1^H NMR	[19]						
	UC	All, AC, IA	Serum	Y, A, O	GC-MS	[20]						
	IBD	AC	Serum	A, O	^1^H NMR	[15]						
Leucine	CD	Unknown	Feces	Y, A, O	^1^H NMR	[25]	Mouse	Colon, serum	>8–24	GC-MS	DSS (A)	[17]
	CD	AC, IA	Feces	A, O	^1^H NMR	[4]						
	UC	AC	Feces	A, O	^1^H NMR	[4]						
	IBD	AC	Serum	A, O	^1^H NMR	[15]						
**Linoleic acid**	CD (ICD)	IA	Feces	Y, A, O	FT-ICR-MS	[27]	Mouse	Colon (distal), cecum	>3–8	UPLC/ToFMS	T-syn deficiency	[21]
Lysine	CD	AC	Plasma	A, O	^1^H NMR	[18]	Mouse	Colon, plasma, liver	>8–24	^1^H NMR	DSS (A)	[13]
	UC	AC	Serum, plasma	A, O	^1^H NMR	[18]	Mouse	Plasma	>3–8	^1^H NMR	*IL10^-/-^*	[26]
	CD, UC	AC	Feces	A, O	^1^H NMR	[4]	Mouse	Feces	>8–24	^1^H NMR	Adoptive	[38]
	CD, UC	Unknown	Feces	Y, A, O	^1^H NMR	[25]					transfer	
**Maleic acid**	UC	All, AC, IA	Serum	Y, A, O	GC-MS	[20]	Mouse	Colon	>8–24	GC-MS	DSS (A)	[17]
Malic acid	CD	All	Serum	Y, A, O	GC-MS	[24]	Mouse	Colon, serum	>8–24	GC-MS	DSS (A)	[17,37]
**Mannose**	CD, UC	AC	Serum, plasma	A, O	^1^H NMR	[18]	Mouse	Serum	>3–8	^1^H NMR	DSS (A)	[16]
Methionine	CD	All	Serum	Y, A, O	GC-MS	[24]	Mouse	Colon	>8–24	GC-MS	DSS (A)	[17]
	UC	AC	Serum	A, O	^1^H NMR	[18]	Mouse	Plasma	>3–8	^1^H NMR	*IL10^-/-^*	[26]
							Mouse	Feces	>8–24	GC-MS	*Winnie*	[29]
**Oleic acid**	CD (ICD)	IA	Feces	Y, A, O	FT-ICR-MS	[27]	Mouse	Feces	>8–24	GC-MS	*Winnie*	[29]
**Phenylacetylglycine**	UC	All	Urine	A	^1^H NMR	[43]	Mouse	Urine	>8–24	NMR	*IL10^-/-^*	[44]
							Mouse	Serum	>24	UPLC-ESI-TOF-MS	H. hepaticus	[45]
							Rat	Urine	?	UPLC-MS/MS, UPLC-ESI-QTOF-MS	TNBS	[31,32]
**Phenylalanine**	CD	AC	Feces	A, O	^1^H NMR	[4]	Mouse	Plasma	>8–24	^1^H NMR	DSS (A)	[13]
	UC	AC	Serum	A, O	^1^H NMR	[14]	Mouse	Colon, serum	>8–24	GC-MS	DSS (A)	[17]
	IBD	AC	Serum	A, O	^1^H NMR	[15]	Mouse	Plasma	>3–24	^1^H NMR	*IL10^-/-^*	[26]
							Mouse	Feces	>8–24	GC-MS	*Winnie*	[29]
							Mouse	Feces	>8–24	^1^H NMR	Adoptive transfer	[38]
Proline	CD	AC	Serum	A, O	^1^H NMR	[18]	Mouse	Colon	>8–24	GC-MS	DSS (A)	[17]
	CD	All	Serum	Y, A, O	GC-MS	[24]						
**Prostaglandin E2**	CD	Unknown	Urine	A	LC-MS	[46]	Rat	Colon	>8–24	LC-MS	DSS (C)	[47]
**Pyruvic acid**	UC	AC	Serum, urine	A, O	^1^H NMR	[18]	Mouse	Plasma	>3–24	^1^H NMR	*IL10^-/-^*	[26]
							Mouse	Feces	>8–24	GC-MS	*Winnie*	[29]
Succinic acid	CD	All	Serum	Y, A, O	GC-MS	[24]	Mouse	Urine	>3–24	GC-MS	*IL10^-/-^*	[40]
							Mouse	Colon	>8–24	GC-MS	DSS (A)	[17]
							Mouse	Plasma	>3–8	^1^H NMR	*IL10^-/-^*	[26]
							Rat	Urine	?	UPLC-MS/MS	TNBS	[32]
**Taurocholic acid**	CD (ICD)	IA	Feces	Y, A, O	FT-ICR-MS	[27]	Mouse	Colon (distal), cecum	>3–8	UPLC/ToFMS	T-syn deficiency	[21]
Threonine	CD, UC	AC	Urine	A, O	^1^H NMR	[18]	Mouse	Colon, serum	>8–24	GC-MS	DSS (A)	[17]
Tryptophan	UC	AC	Urine	A, O	^1^H NMR	[18]	Mouse	Feces	>8–24	^1^H NMR	Adoptive transfer	[38]
	UC	All	Urine	Y	^1^H NMR	[19]	Mouse	Serum	>3–8	^1^H NMR	DSS (A)	[16]
							Mouse	Liver	>8–24	LC-qTOF-MS	DSS (C)	[22]
Tyrosine	CD	AC	Feces	A, O	^1^H NMR	[4]	Mouse	Colon	>8–24	GC-MS	DSS (A)	[17]
	CD (ICD)	IA	Feces	Y, A, O	FT-ICR-MS	[27]	Mouse	Plasma	>3–8	^1^H NMR	*IL10^-/-^*	[26]
							Mouse	Feces	>8–24	^1^H NMR	Adoptive transfer	[38]
**Uracil**	UC	All, AC, IA	Serum	Y, A, O	GC-MS	[20]	Mouse	Urine	>3–24	GC-MS, NMR	*IL10^-/-^*	[33,40,44]
							Mouse	Colon, serum	>8–24	GC-MS	DSS (A)	[17]
**Urea**	UC	All, AC, IA	Serum	Y, A, O	GC-MS	[20]	Mouse	Serum	>8–24	GC-MS	DSS (A)	[37]
**Valine**	CD	Unknown	Feces	Y, A, O	^1^H NMR	[25]	Mouse	Plasma	>8–24	^1^H NMR	DSS (A)	[13]
	CD	AC, IA	Feces	A, O	^1^H NMR	[4]	Mouse	Colon, serum	>8–24	GC-MS	DSS (A)	[17]
	UC	AC	Feces	A, O	^1^H NMR	[4]						
**Xylose**	CD	AC	Urine	A, O	^1^H NMR	[18]	Mouse	Feces	>8–24	GC-MS	*Winnie*	[29]
	UC	AC	Serum	Y, A, O	GC-MS	[20]						

* Metabolites in **bold** are exclusively **increased** in both IBD animal models and IBD patients compared to healthy controls. Disease: CD: Crohn’s disease; IBD: inflammatory bowel disease; ICD: ileal Crohn’s disease; UC: ulcerative colitis. Activity: AC: active; IA: inactive; All: active + inactive. Age groups: Y: very early onset and young; A: adult; O: old. Platform: FT-ICR-MS: Fourier-transform ion cyclotron resonance mass spectrometry; GC-MS: gas chromatography-mass spectrometry; LC-MS: liquid chromatography-mass spectrometry; LC-qTOF-MS: liquid chromatography quadropole time-of-flight mass spectrometry; MRS: magnetic resonance spectroscopy; NMR: nuclear magnetic resonance; SIFT-MS: selected-ion flow-tube mass spectrometry; UPLC-ESI-(q)TOF-MS: ultra performance liquid chromatography electrospray ionization (quadropole) time-of-flight mass spectrometry; UPLC-MS: ultra performance liquid chromatography mass spectrometry; UPLC-MS/MS: ultra performance liquid chromatography tandem mass spectrometry; UPLC/ToFMS: ultra performance liquid chromatography time-of-flight mass spectrometry. Model: (A): acute; ARE: AU-rich elements; (C): chronic; DSS: dextran sodium sulfate; H. hepaticus: Helicobacter hepaticus; IL: interleukin; T-syn: T-synthase; TNBS: 2,4,6-trinitrobenzenesulfonic acid; TNF: tumor necrosis factor; WT: wild-type.

**Table 5 ijms-21-03856-t005:** Metabolites significantly decreased in inflammatory bowel disease (IBD) vs healthy controls in both humans and animals in the systematic review.

	Human Studies						Animal Studies					
Metabolite *	Disease	Activity	Sample Type	Age Group	Platform	References	Species	Sample Type	Age (Weeks)	Platform	Model	References
**4-Cresol sulfate**	CD	All	Urine	Y, A, O	^1^H NMR	[48]	Mouse	Urine	>8–24	^1^H NMR	DSS (A)	[13]
**Acetic acid**	CD	AC	Serum	A, O	^1^H NMR	[18]	Mouse	Serum	>3–8	^1^H NMR	DSS (A)	[16]
	CD	All	Urine	A	^1^H NMR	[43]	Mouse	Plasma	>8–24	^1^H NMR	DSS (A)	[13]
	CD	Unknown	Feces	Y, A, O	^1^H NMR	[25]						
	UC	AC	Serum	A, O	^1^H NMR	[18]						
	UC	AC	Feces	A, O	GC-MS	[49]						
	UC	All	Feces	A	GC-MS	[6]						
	IBD	All	Urine	A, O	NMR	[50]						
Acetylcarnitine	UC	AC	Serum	A, O	^1^H NMR	[18]	Mouse	Spleen	>8–24	LC-qTOF-MS	DSS (C)	[22]
**Acetylglutamic acid**	CD	IA	Feces	Unknown	UPLC-tof-MS	[30]	Mouse	Serum	>24	UPLC-ESI-TOF-MS	H. hepaticus	[45]
**Aconitic acid**	CD, UC	All	Urine	Y	^1^H NMR	[19]	Mouse	Urine	>3–24	GC-MS	*IL10^-/-^*	[33,40]
	UC	AC	Serum	Y, A, O	GC-MS	[20]						
	IBD	All	Urine	A, O	NMR	[50]						
Acylcarnitine	UC	All	Urine	Y	^1^H NMR	[19]	Mouse	Ileum (distal)	>3–24	LC-MS	*TNF* ^ΔARE/WT^	[23]
Alanine	CD	All	Urine	A	^1^H NMR	[43]	Mouse	Serum	>3–8	^1^H NMR	DSS (A)	[16]
	UC	All	Rectum	Y, A, O	GC-MS	[24]	Mouse	Urine	>8–24	^1^H NMR	Adoptive transfer	[38]
	CD, UC	AC	Colonic mucosa	Unknown	Proton MRS	[35]						
	IBD	IA	Urine	A, O	^1^H NMR	[15]						
	IBD	AC	Colonic mucosa	A	^1^H NMR	[36]						
**Aspartic acid**	CD	IA	Feces	A, O	^1^H NMR	[4]	Mouse	Feces	>3–8	^1^H NMR	DSS (A)	[51]
	UC	AC, IA, All	Serum	Y, A, O	GC-MS	[20]						
**Betaine**	CD, UC	AC	Plasma, urine	A, O	^1^H NMR	[18]	Mouse	Serum	>3–8	^1^H NMR	DSS (A)	[16]
							Mouse	Colon	>8–24	NMR (1H, 1C, 1P)	DSS (A)	[42]
**Butanoic acid**	CD, UC	AC	Feces	A, O	GC-MS	[49]	Mouse	Urine	>8–24	^1^H NMR	DSS (A)	[13]
	CD	AC	Feces	A	GC-MS	[52]	Rat	Urine, Feces	?	UPLC-MS/MS	TNBS	[32]
	CD	AC	Feces	A, O	^1^H NMR	[4]						
	CD	Unknown	Feces	Y, A, O	^1^H NMR	[25]						
Carnitine	CD, UC	All	Urine	Y	^1^H NMR	[19]	Mouse	Serum	>3–8	^1^H NMR	DSS (A)	[16]
**Citric acid**	CD, UC	AC	Serum	A, O	^1^H NMR	[18]	Mouse	Serum	>3–8	^1^H NMR	DSS (A)	[16]
	CD, UC	All	Urine	A	^1^H NMR	[43]	Mouse	Plasma	>8–24	UPLC-MS	DSS (A)	[41]
	UC	AC	Urine	A, O	^1^H NMR	[18]	Mouse	Serum	>8–24	GC-MS	DSS (A)	[17]
	UC	All	Rectum	Y, A, O	GC-MS	[24]	Mouse	Urine	>8–24	NMR	*IL10^-/-^*	[44]
	UC	AC, IA, All	Serum	Y, A, O	GC-MS	[20]	Mouse	Serum	>8	UPLC-ESI-TOF-MS	H. hepaticus	[45]
	IBD	AC, IA	Urine	A, O	^1^H NMR	[15]						
	IBD	All	Urine	A, O	NMR	[50]						
Creatine	IBD	AC	Serum	A, O	^1^H NMR	[15]	Mouse	Plasma	>8–24	^1^H NMR	*IL10^-/-^*	[26]
	IBD	All	Urine	A, O	NMR	[50]						
**Dimethylglycine**	CD	All	Urine	A	^1^H NMR	[43]	Mouse	Plasma	0–3, >8–24	^1^H NMR	*IL10^-/-^*	[26]
Fumaric acid	UC	All	Rectum	Y, A, O	GC-MS	[24]	Mouse	Serum	>8–24	GC-MS	DSS (A)	[17]
	UC	AC, IA, all	Serum	Y, A, O	GC-MS	[20]	Mouse	Liver	>8–24	^1^H NMR	DSS (A)	[13]
							Mouse	Serum	>3–8	^1^H NMR	DSS (A)	[16]
							Mouse	Urine	>8–24	NMR	*IL10^-/-^*	[44]
							Mouse	Plasma	0–3	^1^H NMR	*IL10^-/-^*	[26]
Glucose	CD	AC	Plasma	A, O	^1^H NMR	[18]	Mouse	Serum	>3–8	^1^H NMR	DSS (A)	[16]
							Mouse	Plasma, liver	>8–24	^1^H NMR	DSS (A)	[13]
							Mouse	Serum	>8–24	GC-MS	DSS (A)	[37]
							Mouse	Urine	>3–24	GC-MS	*IL10^-/-^*	[33,40]
							Mouse	Plasma	>8–24	^1^H NMR	*IL10^-/-^*	[26]
Glutamic acid	CD, UC	AC	Colonic mucosa	Unknown	Proton MRS	[35]	Mouse	Feces	>3–8	^1^H NMR	DSS (A)	[51]
	CD	IA	Feces	A, O	^1^H NMR	[4]						
	UC	IA, All	Serum	Y, A, O	GC-MS	[20]						
	UC	All	Rectum	Y, A, O	GC-MS	[24]						
	IBD	AC	Colonic mucosa	A	^1^H NMR	[36]						
**Glutamine**	CD	AC	Plasma, urine	A, O	^1^H NMR	[18]	Mouse	Feces	>3–8	^1^H NMR	DSS (A)	[51]
	CD	All	Serum	Y, A, O	GC-MS	[24]	Mouse	Serum	>3–8	^1^H NMR	DSS (A)	[16]
	CD, UC	AC	Colonic mucosa	Unknown	Proton MRS	[35]	Mouse	Colon, serum	>8–24	GC-MS	DSS (A)	[17]
	UC	All	Serum, rectum	Y, A, O	GC-MS	[24]	Mouse	Liver	>8–24	^1^H NMR	DSS (A)	[13]
	UC	AC, IA, All	Serum	Y, A, O	GC-MS	[20]	Mouse	Plasma	>8–24	^1^H NMR	*IL10^-/-^*	[26]
	UC	AC	Serum	A, O	GC-MS	[17]	Mouse	Feces	>8–24	^1^H NMR	Adoptive	[38]
	IBD	AC	Colonic mucosa	A	^1^H NMR	[36]					transfer	
**Glycero-** **phosphocholine**	CD, UC	AC	Colonic mucosa	Unknown	Proton MRS	[35]	Mouse	Colon	>8–24	^1^H NMR	DSS (A)	[13]
	UC	IA	Colonic mucosa	Unknown	Proton MRS	[35]						
	IBD	AC	Colonic mucosa	A	^1^H NMR	[36]						
Glycine	UC	All	Rectum	Y, A, O	GC-MS	[24]	Mouse	Serum	>3–8	^1^H NMR	DSS (A)	[16]
	IBD	IA	Urine	A	^1^H NMR	[15]	Mouse	Serum	>8–24	GC-MS	DSS (A)	[17]
							Mouse	Feces	>8–24	GC-MS	*Winnie*	[29]
**Hippuric acid**	CD	IA	Urine	A, O	^1^H NMR	[53]	Mouse	Urine	>8–24	^1^H NMR	DSS (A)	[13]
	CD, UC	AC	Urine	A, O	^1^H NMR	[18]	Mouse	Serum	>24	UPLC-ESI-TOF-MS	H. hepaticus	[45]
	CD, UC	All	Urine	A	^1^H NMR	[43]						
	CD, UC	All	Urine	Y, A, O	^1^H NMR	[48]						
	CD, UC	All	Urine	Y	^1^H NMR	[19]						
	IBD	AC, IA	Urine	A, O	^1^H NMR	[15]						
	IBD	All	Urine	A, O	NMR	[50]						
**Histidine**	CD, UC	All	Serum	Y, A, O	GC-MS	[24]	Mouse	Serum	>3–8	^1^H NMR	DSS (A)	[16]
	UC	AC, IA, All	Serum	Y, A, O	GC-MS	[20]						
	IBD	AC	Serum	A, O	^1^H NMR	[15]						
	IBD	All	Urine	A, O	NMR	[50]						
**Hypoxanthine**	CD	AC	Urine	A, O	^1^H NMR	[18]	Mouse	Spleen	>8–24	^1^H NMR	DSS (A)	[13]
Inositol	CD, UC	AC	Colonic mucosa	Unknown	Proton MRS	[35]	Mouse	Colon	>8–24	GC-MS	DSS (A)	[37]
	UC	IA	Colonic mucosa	Unknown	Proton MRS	[35]						
	UC	AC, IA, All	Serum	Y, A, O	GC-MS	[20]						
	IBD	AC	Colonic mucosa	A	^1^H NMR	[36]						
**Isocitric acid**	UC	All	Rectum	Y, A, O	GC-MS	[24]	Mouse	Serum	>8–24	GC-MS	DSS (A)	[17]
	UC	AC, IA, All	Serum	Y, A, O	GC-MS	[20]	Mouse	Urine	>3–24	GC-MS	*IL10^-/-^*	[33,40]
Isoleucine	CD, UC	AC	Colonic mucosa	Unknown	Proton MRS	[35]	Mouse	Feces	>8–24	GC-MS	*Winnie*	[29]
	UC	AC, IA, All	Serum	Y, A, O	GC-MS	[20]						
	UC	All	Rectum	Y, A, O	GC-MS	[24]						
Lactic acid	CD	AC	Colonic mucosa	Unknown	Proton MRS	[35]	Mouse	Serum	>3–8	^1^H NMR	DSS (A)	[16]
	UC	AC, IA	Colonic mucosa	Unknown	Proton MRS	[35]						
	IBD	AC	Colonic mucosa	A	NMR	[36]						
Leucine	CD, UC	AC	Colonic mucosa	Unknown	Proton MRS	[35]	Mouse	Plasma	>8–24	^1^H NMR	*IL10^-/-^*	[26]
	UC	All	Rectum	Y, A, O	GC-MS	[24]						
	UC	AC	Plasma	A, O	^1^H NMR	[18]						
Lysine	UC	All	Rectum	Y, A, O	GC-MS	[24]	Mouse	Feces	>3–8	^1^H NMR	DSS (A)	[51]
	UC	All, IA	Serum	Y, A, O	GC-MS	[20]						
	IBD	All	Urine	A, O	NMR	[50]						
Malic acid	UC	AC, IA, All	Serum	Y, A, O	GC-MS	[20]	Mouse	Serum	>8–24	GC-MS	DSS (A)	[17]
	UC	All	Rectum	Y, A, O	GC-MS	[24]						
Methionine	UC	AC, IA, All	Serum	Y, A, O	GC-MS	[20]	Mouse	Serum	>3–8	^1^H NMR	DSS (A)	[16]
	UC	All	Rectum	Y, A, O	GC-MS	[24]	Mouse	Plasma	>8–24	^1^H NMR	*IL10^-/-^*	[26]
**Methylamine**	CD, UC	Unknown	Feces	Y, A, O	^1^H NMR	[25]	Mouse	Urine	>8–24	^1^H NMR	DSS (A)	[13]
	IBD	All	Urine	A, O	NMR	[50]						
Proline	UC	AC, All	Serum	Y, A, O	GC-MS	[20]	Mouse	Serum	>3–8	^1^H NMR	DSS (A)	[16]
	UC	All	Rectum	Y, A, O	GC-MS	[24]	Mouse	Urine	>8-24	^1^H NMR	Adoptive transfer	[38]
**Sebacic acid**	UC	IA	Serum	Y, A, O	GC-MS	[20]	Mouse	Feces	>8–24	GC-MS	*Winnie*	[29]
Succinic acid	CD	AC	Plasma, urine	A, O	^1^H NMR	[18]	Mouse	Serum	>8–24	GC-MS	DSS (A)	[17]
	CD, UC	AC	Colonic mucosa	Unknown	Proton MRS	[35]	Mouse	Urine	>8–24	NMR	*IL10^-/-^*	[44]
	UC	AC, IA, All	Serum	Y, A, O	GC-MS	[20]	Mouse	Urine	>8–24	^1^H NMR	Adoptive transfer	[38]
	UC	AC	Urine	A, O	^1^H NMR	[18]						
	UC	All	Urine	Y	^1^H NMR	[19]						
	UC	All	Rectum tissue	Y, A, O	GC-MS	[24]						
	IBD	AC, IA	Urine	A, O	^1^H NMR	[15]						
	IBD	All	Urine	A, O	NMR	[50]						
**Taurine**	CD, UC	All	Urine	Y	^1^H NMR	[19]	Mouse	Colon, spleen	>8–24	^1^H NMR	DSS (A)	[13]
	CD	AC	Urine	A, O	^1^H NMR	[18]						
	UC	AC, IA, All	Serum	Y, A, O	GC-MS	[20]						
	IBD	AC, IA	Urine	A, O	^1^H NMR	[15]						
	IBD	All	Urine	A, O	NMR	[50]						
Threonine	UC	IA, All	Serum	Y, A, O	GC-MS	[20]	Mouse	Feces	>3–8	^1^H NMR	DSS (A)	[51]
	UC	All	Rectum	Y, A, O	GC-MS	[24]						
**Triglyceride**	UC	All	Plasma	A	LC-MS/MS	[54]	Mouse	Colon (proximal), ileum (distal)	>8–24	^1^H NMR	*TNF* ^ΔARE/WT^	[23]
							Mouse	Liver	>8–24	^1^H NMR	Adoptive transfer	[38]
**Trimethylamine**	CD, UC	Unknown	Feces	Y, A, O	^1^H NMR	[25]	Mouse	Plasma	>8–24	^1^H NMR	*IL10^-/-^*	[26]
Tryptophan	CD, UC	All	Serum	Y, A, O	GC-MS	[24]	Mouse	Plasma	>8–24	UPLC-MS	DSS (A)	[41]
	UC	AC, IA, All	Serum	Y, A, O	GC-MS	[20]	Mouse	Serum	>8–24	GC-MS	DSS (A)	[17]
							Mouse	Plasma	>8–24	LC-MS	*IL10^-/-^*	[40]
Tyrosine	CD	AC	Plasma	A, O	^1^H NMR	[18]	Mouse	Serum	>3–8	^1^H NMR	DSS (A)	[16]
	UC	AC, IA, All	Serum	Y, A, O	GC-MS	[20]	Mouse	Serum	>8–24	GC-MS	DSS (A)	[17]
	UC	AC	Serum, plasma	A, O	^1^H NMR	[18]	Mouse	Plasma	>8–24	UPLC-MS	DSS (A)	[41]
	UC	All	Rectum	Y, A, O	GC-MS	[24]	Mouse	Plasma	>8–24	^1^H NMR	*IL10^-/-^*	[26]
							Mouse	Feces	>8–24	GC-MS	*Winnie*	[29]

* Metabolites in **bold** are exclusively **decreased** in both IBD animal models and IBD patients compared to healthy controls. Disease: CD: Crohn’s disease; IBD: inflammatory bowel disease; UC: ulcerative colitis. Activity: AC: active; IA: inactive; All: active + inactive. Age groups: Y: very early onset and young;, A: adult; O: old. Platform: GC-MS: gas chromatography-mass spectrometry; LC-MS: liquid chromatography-mass spectrometry; LC-MS/MS: liquid chromatography tandem mass spectrometry; LC-qTOF-MS: liquid chromatography quadropole time-of-flight mass spectrometry; MRS: magnetic resonance spectroscopy; NMR: nuclear magnetic resonance; UPLC-ESI-TOF-MS: ultra performance liquid chromatography electrospray ionization time-of-flight mass spectrometry; UPLC-MS: ultra performance liquid chromatography mass spectrometry; UPLC-MS/MS: ultra performance liquid chromatography tandem mass spectrometry; UPLC-tof-MS: ultra performance liquid chromatography time-of-flight mass spectrometry. Model: (A): acute; ARE: AU-rich elements; (C): chronic; DSS: dextran sodium sulfate; H. hepaticus: Helicobacter hepaticus; IL: interleukin; TNBS: 2,4,6-trinitrobenzenesulfonic acid; TNF: tumor necrosis factor; WT: wild-type.

**Table 6 ijms-21-03856-t006:** Overview of the variation in key experimental elements in animal model studies and the number of studies containing the different versions of each element.

Species & Strain *		Model		Analytical Platform		Biological Sample Type		Age Group (Weeks)	
**Mouse**	22	DSS (A)	12	LC-MS **	15	Colon	12	0–3	3
C57BL/6	14	DSS (C)	2	NMR ***	8	Plasma	8	>3–8	15
BALB/c	2	*IL10^-/-^* (C)	6	GC-MS	6	Urine	8	>8–24	19
C57Bl6/N	1	TNBS (A)	3			Serum	7	>24	2
*Winnie*	1	*TNF*^ΔARE/WT^ (C)	1			Feces	4	Not reported	2
ICR	1	T-synthase	1			Liver	4		
CD1	1	deficiency (C)				Spleen	2		
129/SvEv *Rag2*^-/-^	1	H. hepaticus (C)	1			Ileum	1		
129(B6)-*Il10*^tm1Cgn^/J	1	*Winnie*	1			Cecum	1		
129/SvEv	1	(spontaneous) (C)				Small intestine	1		
**Rat**	3	Adoptive	1			Red blood cells	1		
Sprague-Dawley	2	Transfer (C)				Masseter	1		
Fischer 344	1					Longissimus dorsi	1		
**Piglet**	1								

* Species are in bold with strains belonging to each species listed below. ** refers to all variations of this platform: HPLC-MS/MS (high performance liquid chromatography tandem mass spectrometry), LC-MS (liquid chromatography-mass spectrometry), LC-MS/MS (liquid chromatography tandem mass spectrometry), LC-qTOF-MS (liquid chromatography quadropole time-of-flight mass spectrometry), short column LC-MS, UHPLC-MS (ultra high performance liquid chromatography mass spectrometry), UHPLC/MS-MS (ultra high performance liquid chromatography tandem mass spectrometry), UPLC-ESI-TOF-MS (ultra performance liquid chromatography electrospray ionization time-of-flight mass spectrometry), UPLC-ESI-qTOF-MS (ultra performance liquid chromatography electrospray ionization quadropole time-of-flight mass spectrometry), UPLC-MS/MS (ultra performance liquid chromatography tandem mass spectrometry), UPLC-MS (ultra performance liquid chromatography mass spectrometry), UPLC/ToF-MS (ultra performance liquid chromatography time-of-flight mass spectrometry). *** refers to all variations of this platform: 1H-NMR (proton nuclear magnetic resonance), NMR (nuclear magnetic resonance) (^1^H, ^1^C, ^1^P). Note: one study can contain more than one variant of a key experimental element, e.g., both colon and plasma samples. (A): acute; (C): chronic; GC-MS: gas chromatography-mass spectrometry.

**Table 7 ijms-21-03856-t007:** Overview of the variation in key experimental elements in human studies and the number of studies containing the different versions of each element in the systematic review on metabolomics in inflammatory bowel disease (IBD) patients and IBD animal models.

IBD/IBD Subtype		Analytical Platform		Biological Sample Type		Age Group (Years)	
CD	27	NMR *	13	Feces	9	0–1	0
UC	24	GC-MS **	11	Urine	9	>1 and <18	6
IBD	1	LC-MS ***	5	Colon	4	18–60	21
		SIFT-MS	3	Breath	4	60+	13
		ESI-MS	1	Serum	3	Not reported	1
		FT-ICR-MS	1	Plasma	2		
		Proton MRS	1	Ileum	1		
				PBMC Macrophages	1		

* refers to all variations of this platform: ^1^H-NMR (proton nuclear magnetic resonance), NMR (nuclear magnetic resonance). ** refers to all variations of this platform: GC-MS (gas chromatography-mass spectrometry), GC-tof-MS (gas chromatography time-of-flight mass spectrometry). *** refers to all variations of this platform: HPLC-MS (high performance liquid chromatography mass spectrometry), LC-ESI-MS/MS (liquid chromatography electrospray ionization tandem mass spectrometry), LC-MS (liquid chromatography mass spectrometry), LC-MS/MS (liquid chromatography tandem mass spectrometry), UPLC/ToFMS (ultra performance liquid chromatography time-of-flight mass spectrometry). Note: one study can contain more than one variant of a key experimental element, e.g., both colon and plasma samples. CD: Crohn’s disease; ESI-MS: electrospray ionization mass spectrometry; FT-ICR-MS: Fourier-transform ion cyclotron resonance mass spectrometry; IBD: inflammatory bowel disease; MRS: magnetic resonance spectroscopy; SIFT-MS: selected-ion flow-tube mass spectrometry; UC: ulcerative colitis.

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
