# Peer review of "Translational Potential of Metabolomics on Animal Models of Inflammatory Bowel Disease—A Systematic Critical Review"

_ijms, 2020, doi:10.3390/ijms21113856_

Round 1

Reviewer 1 Report

In this review, Kundsen L. and collaborators presents a very systematic analysis of the metabolomic data available in different studies on Inflammatory Bowel disease (IBD) in both human and animal models. They also provide very interesting comparison of the reported metabolites in human and animal models and discuss their translational potential. This analysis was done through comparison across models, age and techniques used in the different studies. It is accompanied by few tables of the main findings to make it more accessible for the reader. In general, this is an important field and advice to non-experts on the many, and sometimes confusing metabolomics data available. This certainly very valuable review that will help to identify metabolites differentiated between healthy and IBD cases which will be appreciated by the IJMS readership and scientists in the field.

A couple of important points to consider though are: 

1) the authors started with more than 560 studies and finally analyzing only less than 10% of the overall literature through very strict filtering. In a way it is good to analyze the best data available but in the other hand excluding 90% of the literature available will lead to loss of a lot of valuable data. 409 records screened and 318 records were excluded with no clear explanation why they were excluded.

2) 33 full-text articles were excluded, 11 wrong outcomes and 10 wrong study design. Can the authors give examples of such wrong outcomes or wrong study design, this is also helpful for the readers to understand what are the criteria of “wrong outcome”.

3) Table 3 indicates 280 increased metabolites in animal studies while in the text line 106 it is indicated as 281.

Author Response

Response to Reviewer 1 Comments
Comments and Suggestions for Authors
In this review, Knudsen L. and collaborators presents a very systematic analysis of the metabolomic data available in different studies on Inflammatory Bowel disease (IBD) in both human and animal models. They also provide very interesting comparison of the reported metabolites in human and animal models and discuss their translational potential. This analysis was done through comparison across models, age and techniques used in the different studies. It is accompanied by few tables of the main findings to make it more accessible for the reader. In general, this is an important field and advice to non-experts on the many, and sometimes confusing metabolomics data available. This certainly very valuable review that will help to identify metabolites differentiated between healthy and IBD cases which will be appreciated by the IJMS readership and scientists in the field.

Thank you very much for the comments and for pointing out that the high number of excluded studies indeed needs to be addressed along with explanations for reasons for exclusion during the full-text screening. Please find our answers and revisions below.

A couple of important points to consider though are:
1) the authors started with more than 560 studies and finally analyzing only less than 10% of the overall literature through very strict filtering. In a way it is good to analyze the best data available but in the other hand excluding 90% of the literature available will lead to loss of a lot of valuable data. 409 records screened and 318 records were excluded with no clear explanation why they were excluded.

Response 1: The following has been added to the manuscript:
- Lines 363-371: In spite of a thorough literature search strategy, many irrelevant studies were still included in the search results. Of the 409 screened studies, 318 studies were excluded already during the abstract screening due to a number of reasons. More than half of the 318 excluded studies were reviews, editorials, or abstracts. A few were in a language other than English or Danish or duplicates not removed by the reference program. Others were on patients with other diseases than IBD (e.g. Clostridium difficile infection or necrotizing enterocolitis) or IBD animals with other conditions aside from IBD. Additionally, several studies focused solely on degradation products from IBD drugs, while some were in fact microbiome or genome-wide association studies with no metabolomics data.

2) 33 full-text articles were excluded, 11 wrong outcomes and 10 wrong study design. Can the authors give examples of such wrong outcomes or wrong study design, this is also helpful for the readers to understand what are the criteria of “wrong outcome”.

Response 2: Examples of wrong outcomes and wrong study design have been added to the manuscript. After reviewing the chosen reasons for exclusion, we believe that two of the studies excluded with “wrong outcomes” are more correctly classified as “wrong study design”. The numbers have been changed accordingly in Figure 1. The following has been added to the manuscript:
- Lines 371-377: During the full-text screening, 10 studies were excluded with the reason “wrong outcomes” and 12 were excluded with the reason “wrong study design. An example of a wrong outcome was prediction performance estimates, i.e. how well metabolomics could discriminate between UC and control without any quantitative data for specific metabolites. A study categorized as having a wrong study design had two separate studies – one with healthy volunteers and one with CD patients, without any comparison of healthy and CD. After the full-text screening, a total of…

3) Table 3 indicates 280 increased metabolites in animal studies while in the text line 106 it is indicated as 281.

Response 3: The number has been corrected to 280.

Reviewer 2 Report

This systematic review from Knudsen et al. is of high interest to the scientific community. Furthermore, the study design is well organized and the analyses are performed with high validity.

I do not have many comments on this brilliant review but would like the authors to discriminate between acute and chronic colitis models in mice. This is detrimental for the conclusion as only the chronic mouse models reflect mechanisms observed in IBD patients.

Author Response

Response to Reviewer 2 Comments

Comments and Suggestions for Authors

This systematic review from Knudsen et al. is of high interest to the scientific community. Furthermore, the study design is well organized and the analyses are performed with high validity.

I do not have many comments on this brilliant review but would like the authors to discriminate between acute and chronic colitis models in mice. This is detrimental for the conclusion as only the chronic mouse models reflect mechanisms observed in IBD patients.

Response: Thank you for your kind comments and for drawing attention to this very important point. The models in all included animal studies have now been allocated as ‘acute’ or ‘chronic’. This information has been added to Table 6 and the column ‘Model’ in Supplementary Table S3.

The following has been added as a footnote to Supplementary Table S3:

  • *TNBS and DSS models have been allocated as ‘acute’ or ‘chronic’ according to the criteria in Wirtz et al. [1] for chemical induction of acute and chronic colitis models. Models were allocated as ‘chronic’ if they were genetically modified or adaptive.

As the DSS model was the only included model induced as both an acute and chronic model, it has been specified in Tables 4 and 5 and Supplementary Tables S8, S9, and S10 whether the metabolites were detected in the acute or chronic version of the DSS model. Specifications regarding the DSS model have been added in the main text as follows:

  • Line 30: “The acute dextran sodium sulfate model appeared as a good model…”
  • Line 135: “…acute DSS mouse model or the TNBS rat model…”
  • Line 221: “…similarity to human studies was observed with the acute DSS mouse model…”
  • Line 222: “…this model was used in almost half of the included animal studies…”
  • Line 223: “However, only five of the acute DSS mouse model studies analyzed serum samples…”
  • Line 226: “…in serum samples from the acute DSS mouse model…”
  • Line 228: “…serum samples from the acute DSS mouse model…”
  • Line 229: “Conversely, the acute DSS mouse model could account for…”
  • Line 231: “…from the acute DSS mouse model as having good translational potential…”
  • Line 270: “…were increased in the acute DSS mouse model.”
  • Line 274: “Interestingly, animal studies using the acute DSS or TNBS models…”
  • Line 276: “…samples from the acute DSS mouse model…”
  • Line 278: “…potential of the acute DSS mouse model.”
  • Line 338: “…the acute DSS model appeared to be the best…”

The following section has also been added to the discussion:

  • Lines 280-289: In the animal studies included in this review, the majority of the studies used models induced as acute models, which is in contrast to the fact that IBD in the human patients is a chronic condition. However, the 68 different metabolites differentiated in both IBD patients and IBD animal models were from both acute and chronic models. This, along with the observation that the acute DSS model had the most similarities with IBD patients in terms of differentiated metabolites, indicates that the specific mechanism operative in the genesis of the inflammation may be of greater importance than whether an animal model is induced as acute or chronic when studying metabolomics in IBD [2]. It should however be noted that none of the existing models truly recapitulates the spontaneous and fluctuating nature of the human disease. The limitations of each model should always be taken into consideration before directly applying experimental findings to the human condition [3].

  1. Wirtz, S.; Popp, V.; Kindermann, M.; Gerlach, K.; Weigmann, B.; Fichtner-Feigl, S.; Neurath, M.F. Chemically induced mouse models of acute and chronic intestinal inflammation. Nature protocols 2017, 12, 1295-1309, doi:10.1038/nprot.2017.044.
  2. Kiesler, P.; Fuss, I.J.; Strober, W. Experimental Models of Inflammatory Bowel Diseases. Cellular and molecular gastroenterology and hepatology 2015, 1, 154-170, doi:10.1016/j.jcmgh.2015.01.006.
  3. Bamias, G.; Arseneau, K.O.; Cominelli, F. Mouse models of inflammatory bowel disease for investigating mucosal immunity in the intestine. Current opinion in gastroenterology 2017, 33, 411-416, doi:10.1097/MOG.0000000000000402.